# Effects of Gluten-Free Diet in Non-Celiac Hashimoto’s Thyroiditis: A Systematic Review and Meta-Analysis

**DOI:** 10.3390/nu17213437

**Published:** 2025-10-31

**Authors:** Edilene Maria Queiroz Araújo, Claubert Radamés Oliveira Coutinho-Lima, André Silva de Sousa, Lana Mércia Santiago de Souza, Helton Estrela Ramos, Bianca de Almeida-Pititto, Graziela De Luca Canto, Virginia Fernandes Moça Trevisani

**Affiliations:** 1Postgraduate Program in Evidence Based Medicine, Federal University of São Paulo (UNIFESP), São Paulo 04021-001, SP, Brazil; anndressousa@gmail.com (A.S.d.S.); graziela.canto@ufsc.br (G.D.L.C.); vfm.trevisani@unifesp.br (V.F.M.T.); 2Nutritional Genomics and Metabolic Dysfunctions Research and Extension Center (GENUT), Department of Life Sciences, State University of Bahia (UNEB), Salvador 41.150-000, BA, Brazil; radamescoutinho@hotmail.com (C.R.O.C.-L.); lanamercia@gmail.com (L.M.S.d.S.); 3Postgraduate Program in Pharmaceutical Sciences (UNEB), Salvador 41.150-000, BA, Brazil; 4Postgraduate Program in Interactive Processes of Organs and Systems, Institute of Health Sciences (ICS), Federal University of Bahia (UFBA), Salvador 40.110-902, BA, Brazil; ramoshelton@gmail.com; 5Health Sciences Center, Federal University of the Reconcavo of Bahia (UFRB), Santo Antônio de Jesus 44.430-622, BA, Brazil; 6Bioregulation Department, Institute of Health Sciences (ICS), Federal University of Bahia (UFBA), Salvador 40.110-100, BA, Brazil; 7Department of Preventive Medicine (UNIFESP), São Paulo 04024-002, SP, Brazil; bapititto@unifesp.br; 8Brazilian Centre for Evidence-Based Research, Federal University of Santa Catarina, Florianopolis 88.040-900, SC, Brazil; 9Department of Medicine, University of Santo Amaro, São Paulo 04743-030, SP, Brazil

**Keywords:** gluten-free, diet, Hashimoto’s thyroiditis, non-celiac disease

## Abstract

**Background/Objectives**: The gluten-free diet (GFD) may be anti-inflammatory in treating Hashimoto’s thyroiditis (HT), but the studies are inconsistent. **Methods**: To determine the effects of the GFD in non-celiac HT, we included randomized controlled trials from the following databases: Cochrane Central, Embase, Lilacs, Medline, Scopus, and Web of Science. The study was registered at Prospero (no. CRD42024566034). The outcomes assessed included free triiodothyronine (fT3), free tetraiodothyronine (fT4), thyroid stimulating hormone (TSH), Anti-thyroid Peroxidase (TPO), anti-thyroglobulin (Tg), C-reactive protein (CRP), body weight (BW), body mass index (BMI) and adverse effects. Sensitivity, subgroup, meta-regression, bias risk, and evidence analyses’ certainty were also assessed. **Results**: Only three studies were meta-analyzed, comprising 110 participants. The pooled data revealed the evidence was very uncertain about the effect of GFD compared to the control group on mean differences (MD) of TSH (MD −0.63 uIU/mL; 95% CI −1.63 to 0.36; *p* = 0.21), fT3 (MD −0.18 pg/mL; 95% CI −0.50 to 0.14; *p* = 0.28), fT4 (MD −0.33 ng/dL; 95% CI −0.89 to 0.23; *p* = 0.24), anti-Tg (MD −10.07 IU/mL; 95% CI −17.73 to −2.42; *p* = 0.010), anti-TPO (MD 76.19 IU/mL; 95% CI 46.86 to 108.51; *p* < 0.00001), CRP (MD −0.12 IU/mL; 95% CI −0.30 to 0.07), BW (MD −1.46 kg; 95% CI −6.70 to 3.77), and BMI (MD −1.80 kg/m2; 95% CI −3.30 to −0.31). The quality of evidence was rated as having serious methodological concerns to extremely serious imprecision. **Conclusions**: The GFD decreased anti-Tg and increased the anti-TPO levels, both significantly. There were no significant results on fT3, fT4, and TSH.

## 1. Introduction

Hashimoto’s thyroiditis (HT), also known as chronic lymphocytic thyroiditis, is an autoimmune thyroid disease first described by Hiroshi Hashimoto in 1912 [1,2]. It is the most common organ-specific autoimmune disease [3], characterized by diffuse lymphocytic infiltration, parenchymal atrophy, fibrosis, and eosinophilic changes in some thyroid follicular cells [1,4,5]. It is the leading cause of hypothyroidism in patients [6,7]. HT is diagnosed by the presence of positive specific Thyroid Antibody Titers (TAT), Anti-thyroid Peroxidase (TPO), anti-thyroglobulin (Tg) [1,2,5], as well as thyroid ultrasound, a complementary method that can be used in the evaluation of thyroid disorders [8]. HT has a general prevalence in adults of 7.5% and can reach 14% depending on the geographical region, 17.5% in women, and 6% in men [9].

At present, HT seems to be related to multiple factors, such as genetic susceptibility [10,11,12,13]; environmental factors, including nutritional factors [14,15]; immune disorders; and cellular and humoral immunity, which play a key role in the development of the disease [8,16]. HT is also known to relate to oxidation redox (OS) [17,18] with advanced glycation end products (AGEs) and the decreased expression of antioxidant paraoxonase (PON-1) as a phenomenon of inflammatory and pro-oxidant imbalance in HT [2,17,18]. Many alternative therapies have emerged to treat the ongoing imbalance in HT, such as antioxidant micronutrient supplementation [7,19,20,21] and specific diets such as GFD [17,18,22,23,24,25,26,27,28,29,30].

HT and celiac disease have the same genetic susceptibility and major histocompatibility complex, class II, DQ, HLA-DQ2 and HLA-DQ8 alleles [10,11]. Therefore, gluten can increase intestinal permeability through the C-X-C Motif Chemokine Receptor 3 (CXCR3) in the intestinal epithelium, promoting the release of zonulin in HT patients [11,31,32,33], including non-celiac individuals [31,34]. Zonulin is a modulator of tight junctions that controls the selective permeability of the intestine. The increase in zonulin contributes to inflammation and the risk of autoimmunity in these susceptible individuals [6,35,36]. The authors found that the increase in plasma zonulin levels in the HT group was statistically significant compared to the control group (*p* < 0.001) and were found to be associated with HT in both univariate and multivariate models (*p* < 0.05) [37]. Thus, it is speculated that the withdrawal of gluten from HT patients without celiac disease may have an anti-inflammatory effect [29,38,39,40,41].

A GFD is a nutritional plan that eliminates gluten, a protein known as prolamin (glutenin and gliadin) found naturally in some grains such as wheat, barley, and rye [25,42,43]. GFD is crucial for individuals with celiac disease [41,44,45], autoimmune conditions [44,46,47], non-celiac gluten sensitivity, or wheat allergy [27,33,34,48,49]. The diet emphasizes naturally gluten-free foods, including vegetables, fruits, fish, meat, dairy products, and certain grains, while avoiding any food source that contains gluten, such as bread, pasta, cereals, and many processed foods [40,42,50].

High-quality evidence supporting the GFD for non-celiac TH patients, as caused, for example, by immune-mediated responses to gluten, is neither robust nor convincing. This lack of robust evidence underscores the importance of our systematic review and meta-analysis, which had the general objective of verifying the effects of the GFD in non-celiac TH individuals on levels of TAT, thyroid hormones, and TSH and evaluating the effects of the GFD on secondary outcomes (C-reactive protein, vitamin D, BW, BMI) in addition to whether the GDF intervention improved the quality of life of these individuals and caused any adverse effects.

## 2. Methods

### 2.1. Study Design

A systematic review protocol based on PRISMA-P [51] was developed and registered in the International Prospective Register of Systematic Reviews (PROSPERO, registration ID: CRD42024566034).

The review followed the Cochrane Handbook for Systematic Reviews [52]. We reported the present systematic review using the Preferred Reporting Items for Systematic Reviews and Meta-Analyses (PRISMA) [51] (see checklist in Appendix A).

### 2.2. Eligibility Criteria

#### 2.2.1. Inclusion Criteria

Research criteria were developed based on the PICOS Acronym (Population, Intervention, Comparison, Outcome, Study) (Table 1): (i) adults and older people diagnosed with HT and non-celiac disease; (ii) those following a gluten-free diet intervention; (iii) valid comparator groups comprising those receiving any gluten dietary intervention, no dietary intervention, or placebo (if all contain gluten); (iv) randomized controlled trials (RCTs), including crossover trials.

#### 2.2.2. Exclusion Criteria

We excluded studies based on the following criteria: (i) pregnancy or lactation; (ii) gluten-reduced diet intervention; (iii) patients with thyroidectomy; (iv) supplementation intervention, such as iron or selenium; (v) interventions with other supporting diets, such as low-carbohydrate diet, Mediterranean diet, hypocaloric diet, or Brazilian cardio-protective diet (DICA/BR diet); (vi) multifactorial interventions, such as lifestyle or physical exercise; (vii) interventions that induced autoimmune thyroid diseases; (viii) patients who started HT drug therapy at the same time as GFD; (ix) patients who changed HT drug therapy during the GFD period; (x) patients with hyperthyroidism or other endocrine disorders.

The comparator group could be any diet containing gluten, such as usual/unusual diets, without intervention, calculated, prescribed, or guided.

### 2.3. Literature Search Strategy

We meticulously developed a highly sensitive search strategy with the invaluable assistance of a librarian. The search strategy was applied to Cochrane Central, Embase (Elsevier), Lilacs (Latin American and Caribbean Health Sciences Literature in the Virtual Health Library), Medline (via PubMed), Scopus (Elsevier), and Web of Science (Elsevier) and was conducted without language or publication period restrictions. The detailed search strategy is presented in the Appendix A. To ensure the robustness of our search strategy, it was tested by Peer Press [53], as shown in Appendix A. We diligently searched the literature up to August 2024 and updated it on 4 February 2025. We set up email alerts to notify us of any new articles listed in databases, ensuring coverage of the latest publications.

Additionally, we searched the platform clinicaltrials.gov for ongoing or unpublished trials. Two researchers (E.M.Q.A. and C.R.O.C.-L.) also carried out an additional search in the gray literature (Google Scholar and ProQuest), hand-searching the reference lists of the included studies and with experts.

### 2.4. Selection Process

During the initial screening, we conducted a preliminary pilot test encompassing two studies according to inclusion and exclusion criteria. All articles were imported from electronic searches into Rayyan [54] and we deleted duplicate studies, ensuring the highest quality of studies for our review.

After that, two investigators (E.M.Q.A. and C.R.O.C.-L.) independently screened the titles and abstracts and performed the study selection in Phase 1. In Phase 2, the same authors (E.M.Q.A. and C.R.O.C.-L.) screened the references based on the full text. Any disagreements were resolved through discussion, or, if required, a third author was consulted (A.S.S.).

### 2.5. Data Collection Process

A data extraction template was formulated using Microsoft Excel, which was tested by two authors (E.M.Q.A. and C.R.O.C.-L.) and comprised two studies. We made some adjustments to the collection instrument during the process. After that, the same two authors (E.M.Q.A. and C.R.O.C.-L.) independently extracted data.

The following items were extracted: study identification, author names, year of publication, study location, participant characteristics (sex, age, weight, BMI), study characteristics (study design, total duration, Hashimoto’s thyroiditis diagnosis, celiac disease exclusion, follow-up, eligibility criteria); type of intervention; duration, frequency, and concentration; and type of outcomes: primary and secondary outcomes were specified and collected, and time points were reported.

We included funding, adverse effects, and notable conflicts of interest for the study authors. Any disagreements were resolved through discussion, or, if required, a third author was consulted (A.S.S.). We contacted authors for unreported outcomes and missing data through three consecutive emails, 15 days apart.

### 2.6. Data Items

There is no established consensus for outcomes in the Core Outcomes Measures in Effectiveness Trials (COMET) [55], so the outcomes chosen were based on guidelines and RCTs. We analyzed the following outcomes in the review but did not use them as a basis for including or excluding studies: primary outcomes: anti-thyroid antibodies (anti-Tg, anti-TPO), thyroid hormones (fT3, fT4), and TSH, measured throughout blood sample collection; secondary outcomes: anthropometric measurements (body weight, BMI), measured with scale and stadiometer; health-related quality of life (we accepted any scales as long as they were validated); diet adherence as described by the authors; adverse effects as defined by the trial authors; and CRP and vitamin D measured throughout blood sample collection.

All the parameters were measured from baseline, before the nutrition intervention, and after the nutrition intervention (change from baseline and final values). Nevertheless, some of them had only baseline parameters. We analyzed only the outcomes listed above for each trial and not all the outcomes reported. We considered the following features in the normal range [56,57]: anti-TPO (<5.61 IU/mL); anti-Tg (<4.11 IU/mL); TSH (0.35–4.94 uIU/mL); fT3 (1.58–3.91 pg/mL); fT4 (0.7–1.48 ng/dL); vitamin D (30–50 ng/mL (≥20–<30 ng/mL = insufficient; <20 ng/mL = deficient)); CRP (<0.1 mg/dL: low risk; 0.1 to 0.3 mg/dL: intermediate risk; >0.3 mg/dL: increased risk); BMI (18.5 to 24.9 kg/m^2^ (adults); 22 to 27 kg/m^2^ (older people)).

### 2.7. Assessment of Risk of Bias in Included Studies

Two authors (E.M.Q.A. and C.R.O.C.-L.) independently assessed the risk of bias using the Cochrane Risk of Bias tool 2 (Rob2), as outlined in the Cochrane Handbook for Systematic Reviews of Interventions [58]. We resolved disagreements by discussing or involving another author (A.S.S.). We assessed the effect of the assignment to the intervention (the intention to treat (ITT) effect), so we performed all risk of bias assessments on this effect, according to the following domains: bias arising from the randomization process, bias due to deviations from intended interventions, bias due to missing outcome data, bias in the measurement of the outcome, and bias in the selection of the reported result. We used the signaling in the RoB2 tool and rated each domain as ‘low risk of bias,’ ‘some concerns’, or ‘high risk of bias.’ The overall risk of bias for the result was the least favorable assessment across the bias domains [52].

We used Excel/Rob2 to generate traffic light plots for each outcome’s domain-level judgments. If more than 10 studies were included in the meta-analysis, a funnel plot was created to explore publication bias. None of the included studies were crossover trials. If crossover trials are included in future updates of this review, a specific bias analysis tool will be used for this type of study.

### 2.8. Synthesis Methods

A narrative synthesis was initially performed, tabulating the literature for each category of all relevant parameters. Data provided by a single study was not meta-analyzed, as it was impossible to compare. We included only the relevant arms, according to PICOS, for studies with more than two groups.

### 2.9. Measures of Treatment Effect

All our outcomes were continuous data. For continuous outcomes collected on the same scale, we pooled the effect using a mean difference (MD) with corresponding standard deviation (SD) and 95% CI; for continuous outcomes collected with different scales, we used a standardized mean difference (SMD) with 95% CI. Data were pulled using a random-effect model, and a forest plot with MD for primary and secondary outcomes was used. Separate analyses were conducted for each outcome and each comparison and we presented numerical data on forest plots using Review Manager (Review Manager—computer program, version 7.2.0, The Cochrane Collaboration, 2024; available at https://revman.cochrane.org/).

#### 2.9.1. Dealing with Missing Data

The study authors were contacted to request unclear numerical data and information (such as randomization and allocation). We did not use imputation because most of the data was available. One author Poblocki et al., 2021 [59] sent his data, which we used for meta-analysis using the generic inverse variance method. We also used the methods recommended to convert the median to the mean [60,61].

#### 2.9.2. Assessment of Heterogeneity

Statistical heterogeneity was assessed using Cochran’s Q test to determine whether heterogeneity was significant (considering a threshold value of *p* < 0.1 to indicate that heterogeneity was statistically significant). In addition, we used Higgins’ I2 over 50% to represent substantial heterogeneity. We visualized and inspected the forest plots for signs of heterogeneity (e.g., non-overlapping CI). If substantial heterogeneity was identified and information was available, subgroup analyses would be carried out according to the characteristics of the population (gender, age group) and the outcome (BMI) [32]. However, we performed a random-effect meta-regression analysis, considering the high risk of bias and gluten concentration (capsule or free consumption).

#### 2.9.3. Sensitivity Analysis

We intended to remove the studies at high risk of bias. However, the studies at a high risk of bias were those with gluten-free consumption.

### 2.10. Assessment of Reporting Bias

The publication of bias would be tested by visual inspection of funnel plots and using Egger’s regression asymmetry test if there were fewer than ten studies in a meta-analysis [34,62].

### 2.11. Assessment of the Certainty of the Evidence

The certainty of the evidence was rated using the GRADE methodology approach (risk of bias, consistency of effect, imprecision, indirectness, and publication bias) for all outcomes by two independent reviewers (E.M.Q.A. and C.R.O.C.-L.). Disagreements were resolved through deliberation or by a third reviewer (A.S.S.). We used the methods and recommendations described in the Cochrane Handbook for Systematic Reviews of Interventions [63] and the GRADE Handbook using GRADEpro GDT software (McMaster University and Evidence Prime, 2025. Available at https://www.gradepro.org/. GRADEpro). We justified all decisions to downgrade the quality of studies using footnotes and made comments to aid the reader’s understanding of the review where necessary. We used four levels to rate the quality of evidence across trials in the GRADE system: very low, low, moderate, and high [63]. We interpreted SMD analysis following the rule of thumb on Cohen’s effect size [64]: 0.2 represents a small effect; 0.5 is a medium effect; 0.8 represents a large effect.

## 3. Results

### 3.1. Study Selection

The flowchart in Figure 1 illustrates the results of the study selection process, as PRISMA recommended [51]. From the 459 records through databases and 979 from other sources, after removing duplicates, we screened 306 titles and abstracts. Then, through full-text review, we checked the eligibility of 13 reports from databases and 7 reports from other sources. In the end, we included four RCTs, but only three contributed data to meta-analyses.

Five study authors were contacted for clarification (because of insufficient data). Two of them answered us [59,65]. Details are available in Characteristics of Excluded Studies (Appendix A).

### 3.2. Study Characteristics

Data are summarized in Table 2 and additional information can be found in the Appendix A. One study was not included in the meta-analysis due to a lack of data on the control group [18] even upon author contact.

**Figure 1 nutrients-17-03437-f001:**
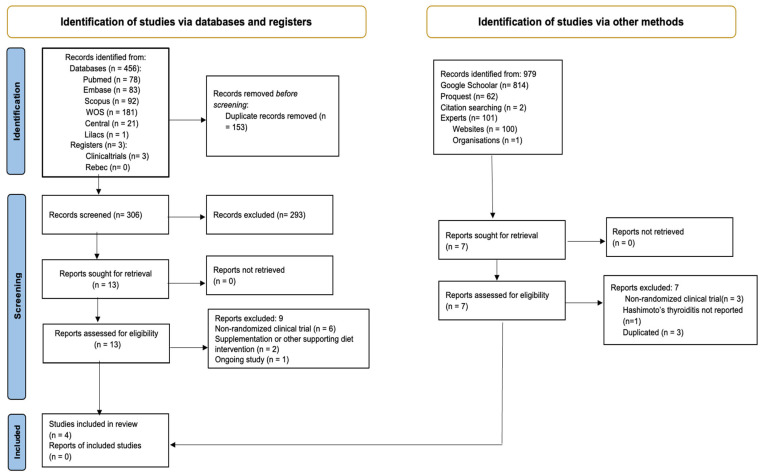
PRISMA flow diagram. Method for the selection of articles for this review.

### 3.3. Population Sociodemographic Characteristics

According to the narrative results, Table 2 shows that the mean age ranged from 34.6 to 43.25 years. Three studies were conducted on women, and only one RCT was conducted on both sexes [18]; the intervention period in the included trials ranged from 1 month to 3 months. The gluten-free diet group received a sample GFD menu or weekly diet according to individual requirements, and the control group did not undergo any modification. Only in one study [65] was there an intervention with the use of capsules for both the gluten-free group (rice starch) and the control group (2 g/day of gluten). Two studies [18,56] had more arms, but data was only extracted from nutritional interventions according to the PICOS: the GFD and control groups [56]. In the trial of Rodziewicz et al., [65], the diets differed; we changed the intervention to the control to better understand the table and meta-analysis standardization. Only one study reported adverse effects [65], as shown in Appendix A.

**Table 2 nutrients-17-03437-t002:** Study characteristics of the published studies included in this systematic review.

*ID*	*Authors*	*Year*	*Country*	*n*	*Study Design*	*Sex*	*Age means (SD or CI)*	*Celiac Disease Exclusion*	*Follow-up (Months)*	*Interventions*		*Outcomes*	*Declaration of Interest*
										*Gluten-Free Diet/Control Group*			
NCT06249074	Rodziewicz L, Szewczyk D, Bryl E [65]	2024	Poland	28	RCT, double-blind study	Women	34.6 (6.3) GG; 36.6 (7.3) PG	Patient was tested for CD: EmA, IgA tTG, total IgA, IgG AGA.	1	Rice starch capsules: placebo + GFD	Gluten (gastrosoluble capsules: 3 cap/2g/day) + GFD	TSH, T3, T4, anti-TPO, anti-Tg, CRP	No conflict
No Information	Poblocki J, et al. [59]	2021	Poland	62	RCT	Women	36.64 (33.66–39.63) GFD; 37.07 (33.83–40.31) CG	Patient was tested for CD: IgA-class antibodies against tTg and total IgA. If necessary: gastroscopy and duodenal biopsy	3	All participants received a sample GFD menu (≤20 mg of gluten per 1 kg natural and processed products)	Did not undergo any modification. They consumed gluten before and during the study.	TSH, T3, T4, anti-TPO, anti-Tg, body weight, BMI	No conflict
NCT05949671	Ulker M, et al. [56]	2023	Turkey	20	RCT, single-blind study	Women	39.05 (7.52)	No symptoms of CD and no diagnosis	3	All participants received a weekly diet according to individual requirements and daily energy needs.	Did not receive any special dietary intervention.	TSH, T3, T4, anti-TPO, anti-Tg, body weight, BMI	No conflict
No Information	Lagana M; Piticchio T; et al. [18]	2025	Italy	30 (20F; 10M)	RCT, single-blind study	Men/women	43.25 * (11.858 **) GFD; 42.25 * (12.434 **) FD (GG).	No diagnosis nor symptoms of CD	3	Patients received information according to individual recommendation by a registered dietitian.	No change in the subjects’ dietary habits.	TSH, T4, anti-TPO, anti-Tg.	No conflict

GG = gluten group; GFD = gluten-free diet; PG = placebo group; CG = control group; FD = free-diet; CD = celiac disease; EmA: anti-endomysial antibodies; tTG = anti-tissue transglutaminase antibodies. Ig: immunoglobulin; AGA: anti-gliadin antibodies; Cap = capsules; BMI = body mass index; CI = confidence interval. M = male; F = female. RCT = randomized clinical trial. * Estimated mean; ** estimated standard deviation. Only the information for this systematic review was collected.

### 3.4. Risk of Bias in Studies

We present the risk of bias assessment for each outcome. We generated traffic light plots in each forest plot of our primary (TSH, fT3, fT4, anti-TG, anti-TPO) and secondary outcomes (CRP, BW, BMI).

Of the three meta-analyzed RCTs, two presented a high overall risk of bias in all outcomes [56,59], while one study showed a low overall risk of bias [65]. Ulker et al.’s (2023) [59] study did not provide sufficient information on the randomization process allocation or how it dealt with missing data, and it presented a high risk of bias due to missing outcome data. Similarly, the study by Poblocki et al. (2021) [59] did not provide sufficient information on the allocation, and the authors had difficulty dealing with missing data and selecting the reported result. Individual studies evaluated secondary data. Only the study containing the CRP [65] presented a low risk of bias.

### 3.5. Effects of Interventions

#### 3.5.1. Thyroid Function Test Outcomes

The evidence is very uncertain about the effect of GFD on TSH levels compared to the control group, and no significant difference is apparent (MD −0.63 uIU/mL; 95% CI −1.63 to 0.36; *p* = 0.21) (Figure 2). Three studies had 110 participants; the certainty of the evidence was very low. We downgraded the certainty of the evidence due to serious methodological concerns (two studies presented a high overall risk of bias) and due to extremely serious imprecision; the CI ranges from a large beneficial effect to a small harmful effect of the intervention (SMD −0.94 μIU/mL, 95% CI −2.25, 0.38) (Appendix A). A high degree of heterogeneity was also detected between the included trials (I2 = 88%, *p* = 0.0003) (Figure 2).

The evidence is very uncertain about the effect of GFD on fT3 levels compared to the control group and there is no significant difference (MD −0.18 pg/mL; 95% CI −0.50 to 0.14; *p* = 0.28); three studies had 110 participants (Figure 3), and the certainty of the evidence was very low. We downgraded the certainty of the evidence due to serious methodological concerns (two studies presented a high overall risk of bias) and due to extremely serious imprecision; the CI ranges from a large beneficial effect to a small harmful effect of the intervention (SMD −0.60 pg/mL, 95% CI −1.64, 0.45) (Appendix A). A high degree of heterogeneity was also detected between the included trials (I2 = 85%, *p* = 0.001) (Figure 3).

The evidence is very uncertain about the effect of GFD on fT4 levels compared to the control group and there is no significant difference (MD −0.33 ng/dL; 95% CI −0.89 to 0.23; *p* = 0.24); three studies had 110 participants (Figure 4) and the certainty of the evidence was very low. We downgraded the certainty of the evidence due to serious methodological concerns (two studies presented a high overall risk of bias) and due to extremely serious imprecision; the CI ranges from a large beneficial effect to a harmful moderate impact of the intervention (SMD −0.78 ng/dL, 95% CI −2.16, 0.59) (Appendix A). A high degree of heterogeneity was also detected between the included trials (I2 = 96%, *p* < 0.00001) (Figure 4).

#### 3.5.2. Immunological Biomarker Outcomes

GFD decreased anti-Tg levels significantly compared to the control, but the evidence is very uncertain (MD −10.07 IU/mL; 95% CI −17.73 to −2.42; *p* = 0.010); three studies had 110 participants (Figure 5), and the certainty of the evidence was very low. We downgraded the certainty of the evidence due to serious methodological concerns (two studies presented a high overall risk of bias) and due to extremely serious imprecision; the CI ranges from a large beneficial effect to a trivial effect or no effect (SMD −0.39 IU/mL, 95% CI −0.84, 0.06) (Appendix A). Moreover, no significant heterogeneity was observed among the included studies (I2 = 0%, *p* < 0.66) (Figure 5).

GFD increased anti-TPO levels significantly compared to the control, but the evidence is very uncertain (MD 76.19 IU/mL; 95% CI 46.86 to 108.51; *p* < 0.00001); three studies had 110 participants (Figure 6), and the certainty of the evidence was very low. We downgraded the certainty of the evidence due to serious methodological concerns (two studies presented a high overall risk of bias) and due to extremely serious imprecision; the CI ranges from a large beneficial effect to a trivial effect or no effect to a large detrimental effect (SMD 0.79 IU/mL, 95% CI −0.09, 1.68) (Appendix A). Additionally, no significant heterogeneity was observed among the included studies (I2 = 0%, *p* = 0.43) (Figure 6).

#### 3.5.3. Inflammation Biomarker Outcomes

The CRP outcome was not meta-analyzed as it was collected in an individual study [65] (Figure 7, Appendix A). The evidence is very uncertain about the effect of GFD on CRP levels considering the comparison with the control group (MD −0.12 IU/mL; 95% CI −0.30 to 0.07); the unique study had 28 participants (Figure 7, Appendix A), and the certainty of the evidence was very low. We downgraded the certainty of the evidence to being of serious methodological concern due to extremely serious imprecision; the CI ranges from a large beneficial effect to a small harmful effect (SMD −0.49 IU/mL, 95% CI −1.25, 0.26) (Appendix A).

#### 3.5.4. Anthropometrics and Quality of Life Outcomes

The body weight outcome was not meta-analyzed as it was collected in an individual study [56] (Figure 8, Appendix A). The evidence is very uncertain about the effect of GFD on body weight levels compared to the control group (MD −1.46 kg; 95% CI −6.70 to 3.77) (Figure 8); this one study had 20 participants (Figure 8, Appendix A); the certainty of the evidence was very low. We downgraded the certainty of the evidence due to serious methodological concerns due to extremely serious imprecision; the CI ranges from a large beneficial effect to a moderate harmful effect (SMD −0.23 kg, 95% CI −1.11, 0.65) (Appendix A).

The BMI outcome was not meta-analyzed as it was collected in an individual study [56] (Figure 9, Appendix A). The evidence is very uncertain about the effect of GFD on BMI levels compared to the control group (MD −1.80 kg/m^2^; 95% CI −3.30 to −0.31); this one study had 20 participants (Figure 9, Appendix A), and the certainty of the evidence was very low. We downgraded the certainty of the evidence due to serious methodological concerns due to extremely serious imprecision; the CI ranges from a large beneficial effect to a trivial effect or no effect (SMD −1.01 kg/m^2^, 95% CI −1.96, −0.07) (Appendix A).

Only one study reported an adverse effect: unfavorable microbiota [65] (Appendix A). The other protocol outcomes (vitamin D, diet adherence, and health-related quality of life) were not found in the included studies.

### 3.6. Sensitivity Analysis

Analysis was carried out considering the studies with a high risk of bias and the concentration of gluten; in the end, the same studies were removed.

### 3.7. Assessment of Reporting Bias

Publication bias was not tested by visual inspection of funnel plots or using Egger’s regression asymmetry test because fewer than ten studies were in the meta-analysis [58].

### 3.8. Protocol Amendments

The only protocol amendment was sensitivity analysis for gluten concentration in the diet. This was the only analysis that could be carried out, even though it had not been planned. The third author L.M.S.d.S. was consulted to resolve any disagreements and was replaced by A.S.S.

## 4. Discussion

The main objective of this systematic review was to verify whether eliminating gluten from the diet would effectively control thyroid hormones and TAT in patients with HT without celiac disease. It is worth noting that no systematic review study has researched the subject with this methodological rigor to date.

The most important findings in this meta-analysis were that GFD decreased the mean difference in anti-Tg and had the opposite effect on anti-TPO, increasing its levels. The effect of GFD on TSH, fT3 and fT4 was not significant and not clinically effective, despite being favorable for this intervention. Regarding the secondary data, CRP, body weight, and BMI, the same was true and so we could not meta-analyze them. For all the results, the certainty of evidence was very low.

HT is the most prevalent autoimmune disease in the world, and many articles have been published on its non-pharmacological treatment, raising many more questions than answers. Scientists are studying the effectiveness of diets in the treatment of HT [66], such as low-calorie diets [26], Mediterranean diets [56], Autoimmune Protocol diets (AIP diet) [28,67], and plant-based diets [68,69], but none arouse as much curiosity in the literature as the gluten-free diet [7,21,27,70].

However, given its growing popularity, knowing the potential nutritional deficiencies resulting from a GFD is essential, especially for those without diseases [41,70,71]. There is aggressive consumer marketing by manufacturers and retailers, scientific research, and the mainstream press about the clinical benefits of avoiding gluten [25]. It can lead people to restrict gluten simply because of the perception that gluten is potentially harmful, and therefore, restriction would only represent a supposedly healthy lifestyle [72]. Although it may seem like a healthy way of life, it can also result in dietary problems and vitamin deficiencies. Gluten-free diets can be high in fat, low in fiber, and deficient in important vitamins and minerals such as calcium, magnesium, and vitamins D and B [25,29,30,38,73,74].

There are few studies in the literature containing data on individuals consuming a GFD without celiac disease and with HT to discuss in this systematic review. Moreover, the studies involve small, short-term cohorts. the authors of [48] conducted the first and only previously published systematic review. The authors found different results from ours: gluten deprivation in HT patients reduced TSH; there was no decrease in TAT; regarding FT3, there was no improvement with gluten withdrawal, and there was a worsening of FT4 with gluten deprivation [48]. The study has methodological concerns highlighted in a recently published commentary [75], possibly explaining the differences between these two reviews. The main one is the mixture of randomized and non-randomized clinical trials in the study by [48].

Still, on the topic of published studies, Ref. [7] (no text available) carried out a clinical trial with 98 women with HT, excluding gluten and taking seleno-methionine (200 μg/Se) in group A and only taking selenium in group B; there was an improvement in thyroid function and autoimmunity after six months. In another 10-week self-control clinical study [67], following an AIP, a modified paleolithic gluten-free diet, no statistically significant changes were observed in thyroid function or antibodies. Still, there was a decrease in systemic and immunological inflammation. For both studies, there was a lack of eligibility criteria for individuals with symptomatic or asymptomatic celiac disease, which makes comparison with this systematic review difficult.

In 2019, authors conducted a non-randomized clinical trial to verify the impact of a gluten-free diet on self-immunization and the function of the thyroid gland in patients with Hashimoto’s disease [27]. The study lasted six months with 34 euthyroid women (aged between 20 and 45). The participants were assigned to two groups: those who followed a gluten-free diet and those who consumed gluten. They concluded that TAT decreased in patients who followed a gluten-free diet. However, as no biopsy of the small intestine was carried out, the patients may have developed subclinical (non-symptomatic) celiac disease. Some of these same authors [21], in 2022, evaluated 91 euthyroid women with HT; 31 of them had non-celiac gluten sensitivity (NGS) and were allocated to group A; the remaining 60 were assigned to group B and did not have NGS. Group A ate a gluten-free diet, and group B ate a gluten diet. Both groups were supplemented with vitamin D. The results showed more pronounced effects of decreased antibodies, anti-TG and anti-TPO, in group B, with the gluten diet. In both studies, the gluten-free diet did not affect thyroid hormones or TSH, similar to this systematic review of randomized clinical trials. In the latter article, it is possible to observe that anti-TPO decreased more in the group that consumed gluten, which is closer to our results. The authors also pointed out that the decrease in antibodies and increase in vitamin D were inversely proportional to gluten intake. The underlying mechanisms for the results found were still unclear. Still, the authors pointed out some explanations, which will be replicated here, and added other hypotheses that could explain our findings about anti-TPO [21].

Most of the randomized controlled clinical trials initially present self-control analyses, i.e., after nutrition intervention compared with the pre-intervention period [18,59,65]. And when they present across the groups, there is no analysis by intervention [56]. This made it difficult to compare this meta-analysis, which evaluated the results of GFD with those of the control group.

Another point to consider is that, according to a recently published article, anti-TPO can indicate adverse health outcomes [76]. Using regression analysis, the authors investigated the association between sociodemographic factors, lifestyle, and mortality risk in 1317 participants in the ELSA-Brazil study. The results showed that anti-TPO was associated with mortality risk and that the determinants of detectable and positive anti-TPO, which were values above the range previously established in the study, were female gender, young age, high BMI, white race/color (self-reported), and low schooling. Regarding female sex, it was the only factor that was associated with persistent anti-TPO, which remained high, above a specific range [76]. When comparing these results with our systematic review, it is similar to the fact that all the participants in the meta-analyzed articles were women and at a young age, which could keep the anti-TPO elevated. What is worrying is that this increase can cause a greater mortality risk, according to the study by [76]. One of the explanations can be found in a prospective cohort study in which 64,714 women and 45,303 men exposed to gluten consumption, estimated by a food frequency questionnaire, were evaluated to assess fatal and non-fatal heart attacks [77]. The authors concluded that long-term gluten consumption was not associated with the risk of coronary heart disease (0.95; 95% CI: 0.88 to 1.02; *p* = 0.29). However, on the contrary, avoiding gluten can reduce the benefits linked to grain consumption, affecting cardiovascular risk. The authors concluded that promoting a gluten-free diet should be discouraged for people without celiac disease.

We considered some hypotheses. Is anti-TPO more sensitive to specific sociodemographic, metabolic, and nutritional characteristics than anti-Tg? In other words, could removing gluten from the diet directly impact HT via anti-TPO, especially in young women?

According to the authors, the implementation of a gluten-free diet, like other elimination diets, is associated with a high risk of nutritional deficiency [20,22,25,29,30,38,78]. In other words, gluten-free products, compared to their traditional counterparts, would have much lower nutritional value. The most common deficiencies in patients following a gluten-free diet are vitamins B and D, calcium, and iron [29,38,79], as well as magnesium, zinc, selenium (Se), and copper [7,19,29,30], alongside high levels of lipids, trans fat, and salt, a high glycemic index, and low concentrations of protein and fiber in the diet [20,25,29,30]. Among the nutrients mentioned, studies show a significant deficit in restrictive diets [20,29,30], but anti-TPO improves with supplementation [7,19,20]. An overview in a systematic review [19] showed that the supplementation reduced anti-TPO and anti-tg levels at 3 and 6 months in a population not medicated with LT4, and the results were significant. However, the adverse effect was large compared to the non-supplemented group (RR: 2.93; CI = 0.93 to 6.11). However, the level of evidence for all the results was low. These effects of Se may be related to its antioxidant capacity, mainly due to GPx3, a strictly Se-dependent enzyme present in the thyroid, which reduces the oxidative effects of excess H_2_O_2_ produced in the thyrocyte [30].

Another nutrient that may be lacking in the gluten-free diet is vitamin D [29,38,79], in addition to a lack of unsaturated fatty acids, iron, and calcium [20,29], which would decrease the intestinal absorption of that fat-soluble vitamin [29,74]. The mechanisms for this may be because T and B lymphocytes, responsible for immune tolerance in TH [30], have receptors for vitamin D, through which they can produce anti-inflammatory cytokines [80]. Lower amounts of vitamin D would aggravate the inflammatory condition in gluten-free patients [29,30,74]. Patients with HT have high serum levels of advanced glycation end products (AGE) and reduced activity of the main thyroid antioxidant enzymes compared to controls, indicating a condition of oxidative stress [17]. That is why it is essential to eat a diet balanced in antioxidant micronutrients, selenium, zinc, iron, and vitamins C, A, E, and D. A recent study proved the ineffectiveness of GFD in improving the redox balance in TH patients without celiac disease [18]. Moreover, in another study, also published recently, with GFD and omega 3 (eicosapentaenoic acid/EPA and docosahexaenoic acid/DHA), the authors observed an increase in anti-inflammatory mediators. However, it was attributed much more to the essential fatty acid than the lack of gluten in the diet [22].

Still, concerning diet quality, GFD significantly increases the consumption of rice products, which can contain heavy metals such as arsenic, mercury, cadmium, and lead [73]. Clinical studies and NHANES-based assessments have indicated that people following a GFD have higher blood/urinary levels of arsenic and mercury, which suggests that the GFD and not the disease is responsible [73]. In addition, these processed foods contain more fats and carbohydrates than their traditional counterparts [25,40,70,74,81] and often more kilocalories, leading to insulin resistance, diabetes, and obesity [26,77]. Ostrowska et al. (2021) [26] evaluated 100 women aged 18–65 previously diagnosed with HT and obesity and taking L-thyroxine. Fifty were randomly assigned to group A (kilocalorie-restricted diet and reduction in sensitive foods, after food testing) and fifty to group B (kilocalorie restriction only). All the diets in both groups were designed by qualified nutritionists, with the same distribution of macronutrients and meeting the daily needs of micro- and macronutrients. All participants received 200 mcg of l-seleno-methionine/day and 30 mg of zinc gluconate/day throughout the study period. The parameters evaluated (BMI, TSH, fT3, fT4, anti-TPO, and anti-TG) were significant for diet A, which also included control of gluten intake, compared to control group B. This study also showed a positive correlation between anti-TPO levels and BMI (*p* = 0.023) but not between anti-TPO and decreased body fat percentage (*p* = 0.156). Corroborating this result, obesity was significantly correlated with positive anti-TPO (RR = 1.93, 95% CI 1.31–2.85, *p* = 0.001) and not with anti-Tg in a meta-analysis of 22 studies [35]. In other words, anti-TPO seems more sensitive to the anthropometric profile than anti-Tg.

Another aspect mentioned in the article by Rodziewicz et al. (2024) [65] was the intestinal microbiota, which is quite controversial; it is not known whether its alteration is present before the onset of HT or whether it reflects the effects of the disease itself [82,83,84]. However, the intestinal microbiota influences both the activity of the immune system and thyroid function [35,83,85,86,87,88,89,90]. In addition, gut microbiota affects the availability of micronutrients essential for thyroid function [82,84]. Zafeiropoulou et al. (2020) [39] observed a decrease in short-chain fatty acid (SCFA) concentration after six months of GFD. It is worth noting that SCFA modulates the autoimmune response, maintains intestinal homeostasis, and has anti-inflammatory properties, especially butyrate, which seems to be associated with an increase in the colonic population of Treg lymphocytes, capable of suppressing the auto-reactive immune response [36,91]. In addition, a recent study [92] associated the composition of the microbiota with the anti-TPO of euthyroid participants. The results suggested no robust difference in the gut microbiome between individuals with or without anti-TPO in terms of alpha and beta diversity. However, several taxa with nominal significance related to the presence of TPOAb were identified.

Based on these findings in the literature, it is possible to perform a comparative analysis with the studies included in this systematic review. The RCTs included had eligibility criteria (Appendix A) ranging from normal BMI [65] to no information [59] to grade I obesity [56], and only one study assessed BMI [56], so it was not possible to meta-analyze the data. Therefore, it is not safe to say that GFD reduces BMI; on the contrary, energy consumption is much higher in those who follow GFD [70,77]. Only one study reported on energy intake precautions [56], despite having the control and guidance of specialized professionals for the gluten-free diet [56,59], and there was no information on micronutrient orientation, especially antioxidants, or on the consumption of processed foods. In addition, the sample comprised women and young people who were in situations that may have been responsible for raising and maintaining anti-TPO.

As for adverse effects, there were no reports in the studies included in this SR; only Poblocki et al. (2021) [59] described unfavorable microbiota results following the use of GFD. This lack of adverse effects, not reported in the studies, is not in line with Di Sabatino et al. (2015) [50], who conducted a randomized, placebo-controlled, crossover study with 61 adults without celiac disease or wheat allergy to investigate intestinal and extra-intestinal symptoms related to gluten consumption. The authors observed that, according to the per-protocol analysis of the 59 patients who completed the study after one week of intervention, gluten intake significantly increased general symptoms compared to the gluten-free placebo (*p* = 0.034). Abdominal bloating (*p* = 0.04) and pain (*p* = 0.047) as intestinal symptoms and foggy mind (*p* = 0.019), depression (*p* = 0.020), and aphthous stomatitis (*p* = 0.025) among extra-intestinal symptoms were significantly more severe when subjects were given gluten. In this study, patients who dropped out of the study were not evaluated, which presents a significant bias, as only people with gluten sensitivity who were symptomatic and non-celiac, i.e., those who would benefit from the gluten-free diet, may have remained.

This systematic review, the first of its kind in terms of methodological rigor, has certain limitations that should be considered when interpreting the findings. The short follow-ups and small numbers of participants may limit the generalizability of the results. Additionally, the subjects meta-analyzed were all females, making it impossible to address the effects of the GFD on TAT and thyroid function tests in male subjects. Furthermore, the studies did not assess adherence to the gluten-free diet, which introduces the risk of patients not adhering to restrictive diets. The potential impact of these limitations on future research is significant, underscoring the need for more rigorous studies in this area.

## 5. Conclusions

The GFD, compared to the control group diet, did not significantly influence TSH and thyroid hormones, fT3 and fT4; as for TAT, GFD decreased anti-Tg and increased the anti-TPO levels, both significantly. The certainty of evidence was very low for all outcomes, highlighting the need for more rigorous studies in this area.

These findings underscore the urgent need for more rigorous studies in this area of research. We need more randomized clinical trials with greater control of the GFD, and most importantly, we need more precise research methods. Adhering to the research method guidelines for randomized clinical trials is crucial to avoid high risk of bias and improve the certainty of evidence.

## Figures and Tables

**Figure 2 nutrients-17-03437-f002:**
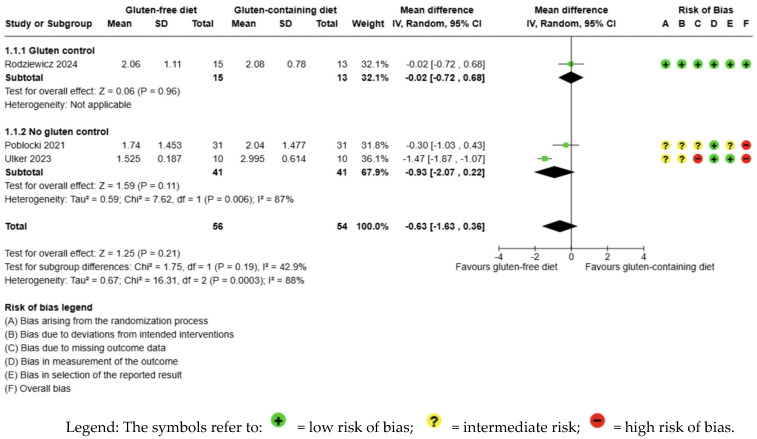
Forest plot: effect of GFD compared to control group on TSH in Hashimoto’s thyroiditis patients with non-celiac disease [56,59,65].

**Figure 3 nutrients-17-03437-f003:**
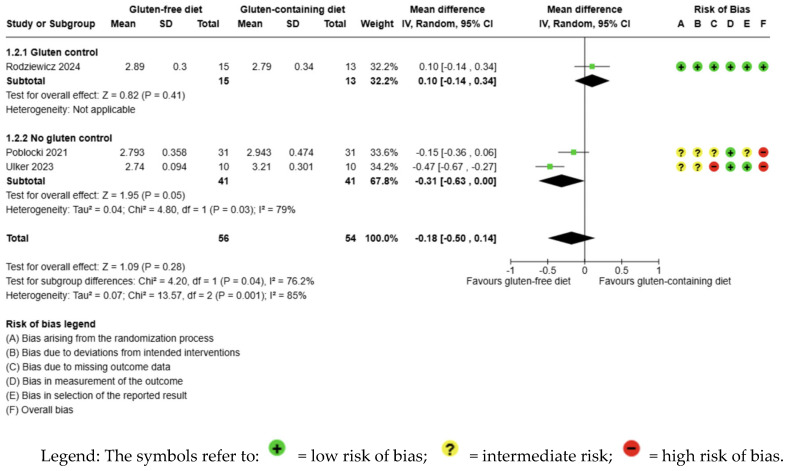
Forest plot: effect of GFD on FT3 compared to the control group in Hashimoto’s thyroiditis patients with non-celiac disease [56,59,65].

**Figure 4 nutrients-17-03437-f004:**
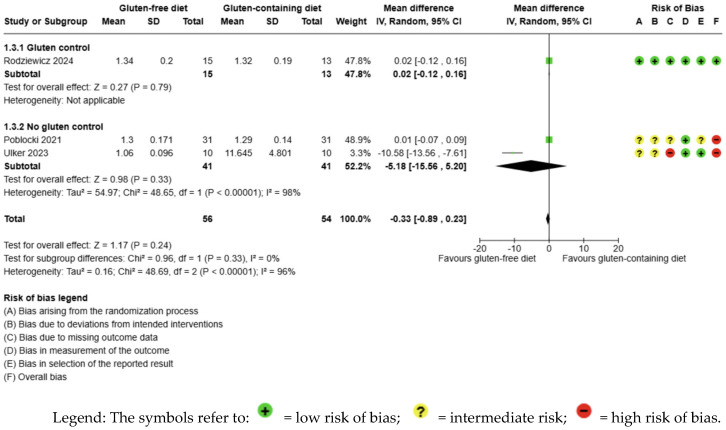
Forest plot: effect of GFD on FT4 compared to the control group in Hashimoto’s thyroiditis patients with non-celiac disease [56,59,65].

**Figure 5 nutrients-17-03437-f005:**
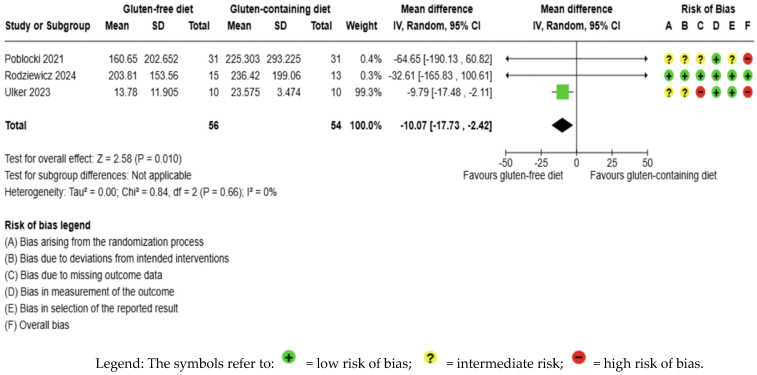
Forest plot: effect of GFD on anti-TG compared to the control group in Hashimoto’s thyroiditis patients with non-celiac disease [56,59,65].

**Figure 6 nutrients-17-03437-f006:**
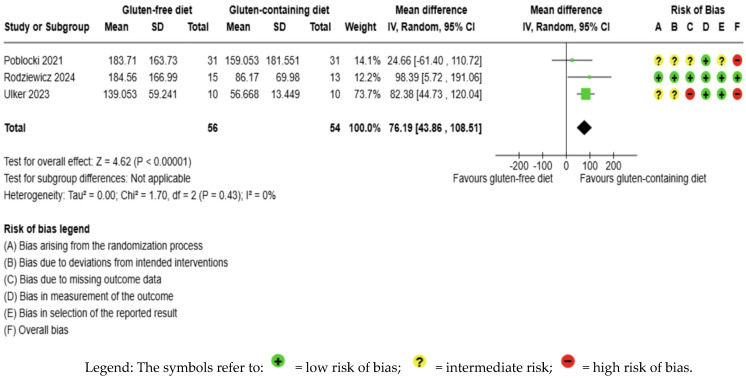
Forest plot: effect of GFD on anti-TPO compared to the control group in Hashimoto’s thyroiditis patients with non-celiac disease [56,59,65].

**Figure 7 nutrients-17-03437-f007:**
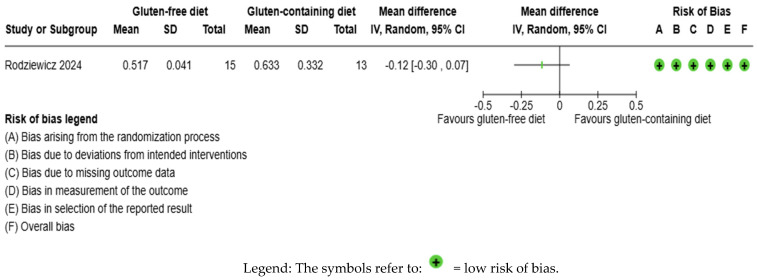
Forest plot: effect of GFD compared to control group on CRP in Hashimoto’s thyroiditis patients with non-celiac disease [65].

**Figure 8 nutrients-17-03437-f008:**
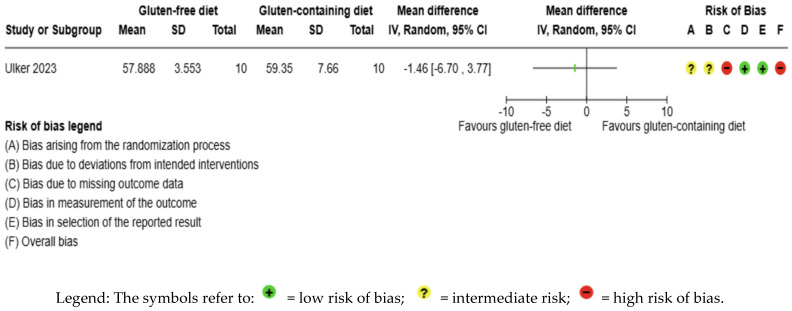
Forest plot: effect of GFD compared to control group on body weight in Hashimoto’s thyroiditis patients with non-celiac disease [56].

**Figure 9 nutrients-17-03437-f009:**
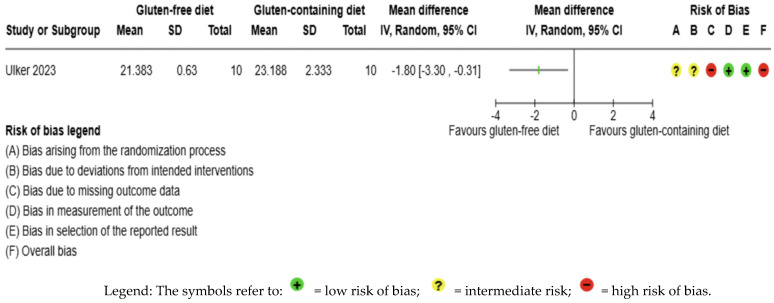
Forest plot: effect of GFD compared to control group on BMI in Hashimoto’s thyroiditis patients with non-celiac disease [65].

**Table 1 nutrients-17-03437-t001:** PICOS criteria for studies’ inclusion in the systematic review.

P = Adults and older people diagnosed with Hashimoto’s Thyroiditis and non-celiac disease
I = Gluten-free diet
C = Any gluten dietary intervention; no dietary intervention; placebo (as long as all of them contain gluten)
O = Primary outcomes: serum levels of thyroid hormones (fT3, fT4); TSH; anti-thyroid antibodies (anti-TPO and anti-Tg). Secondary outcomes: serum levels of CRP and vitamin D; adverse effects; anthropometric measurements (body weight, BMI); diet adherence; and health-related quality of life
S = Randomized controlled trials, including crossover trials

## Data Availability

The data presented in this study are available on request from the corresponding author.

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
