# Peer review of "Effects of Gluten-Free Diet in Non-Celiac Hashimoto’s Thyroiditis: A Systematic Review and Meta-Analysis"

_nutrients, 2025, doi:10.3390/nu17213437_

Round 1
Reviewer 1 Report
Comments and Suggestions for Authors
The systematic review and mata-analysis about studies evaluating the effects of a GFD in non-CD HT subjects by Queiroz Araujo et al is well done, well structured and the study design is well descripted by authors. Statistics has been conducted with clarity and method; both the inclusion and exclusion criteria are well established. The Discussion takes into account the best literature in the field. However, in my opinion, this study does not shed light on conflicting results reporting in the literature and it is not conclusive.
Author Response
Comments 1:
However, in my opinion, this study does not shed light on conflicting results reporting in the literature and it is not conclusive.
R. Thank you for your comments. We investigated all the subjects published up to that exact moment, and only a few dealt exclusively with ANTI-TPO. In other words, our research is unprecedented in relation to this unique data in the literature, and to date, no article has been published on this topic with this methodological rigor. This explains the difficulty in finding articles on the same topic and method to discuss. As for not being conclusive, this happens when there are no original articles in the literature that meet the eligibility criteria and/or have low evidence. However, the review cannot be penalized for this; on the contrary, it shows the fragility of scientific research worldwide.

Reviewer 2 Report
Comments and Suggestions for Authors
This study uses meta-analysis to discuss the impact of a gluten-free diet on non-celiac Hashimoto's thyroiditis.
Introduction:
The discussion of HT is somewhat lengthy and overlooks the connection between GFD and HT.
Note on abbreviations: Some abbreviations lack definition upon first use. For example, "HT" appears in the main text without prior definition, and "Hashimoto's thyroiditis" is not explicitly abbreviated.
Materials and Methods:
What were your exclusion criteria for selecting articles (e.g., excluding reviews, opinion pieces)? I only saw the removal of duplicates.
This study doesn't involve physical "materials," correct? Please change the chapter title accordingly (e.g., just "Methods").
Line 309 (Figures and Tables): Ensure all figures are referenced in the text and provide clear explanations in the captions. For instance, Figure 2 is not referenced here.
Language: I find the language somewhat problematic. Sentences starting with "we" are excessive. Scientific papers should focus on results rather than emphasizing the authors' actions. (Acknowledging this may be subjective, and the author may choose to disregard this point).
Section 3.5: The presentation of results in this section is very dull. Please shorten these uninformative sentences.
Figures: All figures appear stretched. Please ensure they are displayed with normal scaling/aspect ratio.
Chapter Headings: Some headings in Chapters 2 and 3 are too similar. Please revise them to show clearer distinction.
Lines 479-481: The meaning of these two sentences is repetitive.
Discussion of Other Studies: When discussing other authors' research, the focus should be on their findings. Elaborating too much on their methodologies makes these sentences seem verbose.
Although the certainty of evidence in this paper is low, I believe it is still necessary to highlight the potential guidance this study offers regarding gluten-free diets and suggest which nutrients could be supplemented when choosing a gluten-free diet.
Comments on the Quality of English LanguageSee comments
Author Response
Comments 1:
Introduction:
The discussion of HT is somewhat lengthy and overlooks the connection between GFD and HT.
R. The discussion on HT is somewhat lengthy (we can exclude the sections in red, lines 61-63 and 71-73) and ignores the connection between GFD and HT. (This connection is reported between lines 74 and 87).
Note on abbreviations: Some abbreviations lack definition upon first use. For example, "HT" appears in the main text without prior definition, and "Hashimoto's thyroiditis" is not explicitly abbreviated.
R. Some abbreviations lack definition upon first use. For example, “HT” appears in the main text without prior definition, and “Hashimoto's thyroiditis” is not explicitly abbreviated. (Included in line L48)
Materials and Methods:
What were your exclusion criteria for selecting articles (e.g., excluding reviews, opinion pieces)? I only saw the removal of duplicates.
R. We used the acronym PICOS to describe the inclusion criteria (item 2.2.1), describing the only types of studies included, such as (iv) Randomized Controlled Trials (RCTs), including crossover trials. We thought it would be unnecessary to describe the types of studies excluded from the review (item 2.2.2).
This study doesn't involve physical "materials," correct? Please change the chapter title accordingly (e.g., just "Methods").
R. Done.
Results:
Line 309 (Figures and Tables): Ensure all figures are referenced in the text and provide clear explanations in the captions. For instance, Figure 2 is not referenced here.
R. The excerpt from line 309 is part of the topic “3.4 Risk of bias in studies,” in which we simply list/present all the outcomes of our study that will be presented and discussed in the following topic. All figures are presented and discussed in the topic “3.5 Effects of interventions,” and for this reason, Figure 2 is presented in line 326 of subtopic 3.5.1 Thyroid function test outcomes.
Language: I find the language somewhat problematic. Sentences starting with "we" are excessive. Scientific papers should focus on results rather than emphasizing the authors' actions. (Acknowledging this may be subjective, and the author may choose to disregard this point).
R. Excelent suggestion. Done.
Section 3.5: The presentation of results in this section is very dull. Please shorten these uninformative sentences.
R. Thank you for your contribution. These excerpts are part of the explanations about the levels of evidence we attribute to each of the outcomes analyzed, and we the review following was conducted by the Cochrane Handbook for Systematic Reviews (52), Line 100.
Figures: All figures appear stretched. Please ensure they are displayed with normal scaling/aspect ratio.
R. Thank you. Done.
Chapter Headings: Some headings in Chapters 2 and 3 are too similar. Please revise them to show clearer distinction.
R. We have revised them, however, the chapter titles in accordance with the recommendations of the PRISMA Checklist, 2021, for systematic reviews. For this reason, they are titled as follows.
Lines 479-481: The meaning of these two sentences is repetitive.
R. Excluding the section: An imbalance in a gluten-free diet can result in nutritional deficiencies
Discussion
Discussion of Other Studies: When discussing other authors' research, the focus should be on their findings. Elaborating too much on their methodologies makes these sentences seem verbose.
R. Thank you for the suggestion, but it would be important for reviewers to indicate which paragraphs or items they detected these situations in so that we can review them, as discussing methods is part of a systematic review that follows a rigorous standard, such as this one, which, despite not being published in Cochrane, follows its standard.
Although the certainty of evidence in this paper is low, I believe it is still necessary to highlight the potential guidance this study offers regarding gluten-free diets and suggest which nutrients could be supplemented when choosing a gluten-free diet.
R. Thank you for your feedback. First, we remind reviewers that a systematic review does not suggest or recommend the consumption of any product. It only provides results. It is the guidelines that make suggestions. Secondly, there was no emphasis on a gluten-free diet in the treatment of HT. Contrary to popular belief, removing gluten did not improve the signs and symptoms of Hashimoto's thyroiditis, so it would not be possible to suggest supplementation for the general public either.
Round 2
Reviewer 2 Report
Comments and Suggestions for Authors
The manuscript is approved